# A Practical Method for Amino Acid Analysis by LC-MS Using Precolumn Derivatization with Urea

**DOI:** 10.3390/ijms24087332

**Published:** 2023-04-15

**Authors:** Runjin Zhao, Biling Huang, Gang Lu, Songsen Fu, Jianxi Ying, Yufen Zhao

**Affiliations:** 1Institute of Drug Discovery Technology, Ningbo University, Ningbo 315211, China; cpz25cfmpszz@163.com (R.Z.); huangbiling@nbu.edu.cn (B.H.); lgnbdx@163.com (G.L.); zhaoyufen@nbu.edu.cn (Y.Z.); 2Qian Xuesen Collaborative Research Center of Astrochemistry and Space Life Sciences, Ningbo University, Ningbo 315211, China; 3Department of Chemical Biology, College of Chemistry and Chemical Engineering, Xiamen University, Xiamen 361005, China; 4Key Lab of Bioorganic Phosphorus Chemistry & Chemical Biology, Department of Chemistry, Tsinghua University, Beijing 100084, China

**Keywords:** amino acids, quantitative analysis, derivatization, urea, LC-MS

## Abstract

Amino acid (AA) analysis is important in biochemistry, food science, and clinical medicine. However, due to intrinsic limitations, AAs usually require derivatization to improve their separation and determination. Here, we present a liquid chromatography-mass spectrometry (LC-MS) method for the derivatization of AAs using the simple agent urea. The reactions proceed quantitatively under a wide range of conditions without any pretreatment steps. Urea-derivatized products (carbamoyl amino acids) of 20 AAs exhibit better separation on reversed-phase columns and increased response in a UV detector compared to underivatized ones. We applied this approach to AA analysis in complex samples using a cell culture media as a model, and it showed potential for the determination of oligopeptides. This fast, simple, and inexpensive method should be useful for AA analysis in complex samples.

## 1. Introduction

Amino acids (AAs) are the building blocks of proteins and play fundamental roles in many essential biological processes [1,2]. There are two main types of AAs: those that are free and found in food and bodily fluids, and those that are bound and found in peptides and proteins. In addition to serving as the building blocks for cellular metabolism, AAs constitute a significant class of neuroactive substances that are intricately related to a variety of physiological processes and diseases [3]. AAs are essential to human life, and the analysis of AAs has long been a crucial area of study. Therefore, the qualitative and quantitative analysis of AAs in matrices such as biological samples and foods is important in the research of protein chemistry, food science, clinical medicine, and other fields.

Generally, the analysis of AAs from a complex sample undergoes AA separation and AA determination. The AA detectors are mainly based on signal collection from ultraviolet (UV) absorption, fluorescence (FL) emission, mass spectrometry (MS), nuclear magnetic resonance (NMR), and electrochemistry. Of them, NMR does not require complex sample preparation [4,5] but suffers from limited sensitivity. Approaches for AA separation mainly include chromatography and capillary electrophoresis (CE) [6]. A set of chromatographic techniques, including thin-layer chromatography (TLC) [7], liquid chromatography (LC) [8], and gas chromatography (GC) [9] have been employed for AA separation. In recent years, numerous HPLC methods capable of UV/FL detection have been developed for the analysis of AAs [10,11,12]. However, because these detectors are non-specific by-products generated in the derivatization process they may impede the analysis when detecting derivatized AAs. Thanks to the rapid development of mass spectrometry, liquid chromatography–mass spectrometry (LC-MS) has become an essential analytical tool for separating AAs [13]. The structure of the analyte can be determined using MS, making the detection of AAs more reliable. As a detector in HPLC, MS plays an essential role not only in protein analysis but also in metabolite analysis. An advantage of adopting LC-MS system is its selectivity, which may differentiate analytes not only by their retention time but also by their m/z (mass-to-charge ratio) value. Especially with the introduction of MS-acquiring strategies, such as multiple reaction monitoring (MRM) [14,15,16] and selected reaction monitoring (SRM) [17,18], the throughput and quantitative accuracy are further improved.

Given that most AAs are incapable of UV absorption or fluorescence emission and are of high polarity, chemical derivatization is frequently utilized in most AA analysis procedures. Many derivatization reagents have been developed, such as 5-(dimethylamino) naphthalene-1-sulfonyl chloride (dansyl-Cl) [19], O-phthalaldehyde (OPA) [20,21], 7-fluoro-4-nitorobenzo-2oxa-1,3-diazole (NBDF) [22], 9-fluorenylmethylchloroformate (FMOC-Cl) [23], 6-amino-quinolyl-N-hydroxysuccinimidyl carbamate (AQC) [24], p-N, N, N-trimethylammonioanilyl N′-hydroxysuccinimidyl carbamate iodide (TAHS) [25], and phenylisothiocyanate (PITC) [26]. Of these reagents, only AQC, FMOC-Cl, and PITC can react with primary and secondary AAs simultaneously. However, hydrolysis products of AQC and FMOC-Cl may interfere with the detection, while the derivatives of PITC are unstable. Moreover, PITC must be removed from a sample to avoid column contamination. Partial AA derivatives from OPA are also unstable. All the reagents mentioned above need an additional buffer (pH 8~9) for efficient derivatization conversion and are not stable in aqueous environments. Overall, there is still significant scope for developing new derivatization reagents for AA analysis.

Here, we present a method for derivatizing AAs with urea, based on LC-MS detection. Inspired by previous studies [27], we optimized the experimental conditions for the reaction between urea and AAs to generate carbamoyl amino acids (CAAs) in an aqueous solution without any pretreatment. The resulting CAAs from both standard AAs and complex samples show better performance in chromatographic separation and quantitative determination. Details are described below.

## 2. Results and Discussion

### 2.1. Feasibility of Urea Derivatization for Amino Acid Analysis

The reaction of urea and amino acids (AAs) generates carbamoyl amino acids (CAA) along with the release of NH_3_ (Figure 1a). To test the feasibility of urea as an AA derivatization reagent, we chose two primary AAs, alanine (Ala) and phenylalanine (Phe), as model AAs for investigation. The short retention time on reversed-phase columns and low UV response of Ala hinder its detection in conventional HPLC, while Phe is less polar and contains a benzene moiety with UV absorption. These two AAs represent the characteristics of most AAs in the analysis, which can evaluate the effectiveness of the derivatization method well. Stock solutions of urea and Ala (or Phe) were mixed 1:1 and incubated at 60 °C [27,28]. This original condition was initially used to confirm whether the target product CAA could be generated. The corresponding urea-derivatized CAA products were detected in the above reaction by LC-MS (Figure 1b and Appendix A), and MS2 analysis confirmed the reaction products as CAAs (Figure 1c and Appendix A). The quantitative analysis gave the conversion of Ala and Phe as 98.01% and 99.69%, respectively (Appendix A). The conversion rate was calculated by the peak area of EIC, and Equation (1) was used for the calculation.
(1)$$AA\;conversion\;rate=\frac{EIC\left(control\right)-EIC(residual\;AA)}{EIC\left(control\right)}\times 100\%$$

The retention times of Ala and Phe on the reversed-phase C18 column were increased by 1.62 min and 2.59 min after derivatization, respectively, which was helpful for the separation and detection of AAs. In addition, the UV intensities of carbamoyl-Ala and carbamoyl-Phe were 3.29 and 1.26 times that of Ala and Phe, respectively (Appendix A), which enhanced the detection sensitivity of AAs. Taken together, these results show the potential of urea derivatization for AA detection, especially for AAs such as Ala, which is highly polar and has a low UV response. Then, the detection limits and linearity of urea-derivatized AAs were evaluated in the selected ion-monitoring mode (Table 1). The detection limits of Ala and Phe derivatized with urea were 58.6 nM and 11.64 nM, respectively, which are lower than that of Ala and Phe. Additionally, the AA derivatization products with different concentrations can obtain a good linear relationship, which proves that urea is a reliable and promising derivatization reagent for quantitative AA analysis.

### 2.2. Method Optimization

Since the feasibility of this method for qualitative and quantitative AA analysis has been demonstrated, we further refined the experimental conditions to obtain the optimal parameters for the derivatization reaction. First, the appropriate pH conditions were investigated. The reaction mixtures were incubated at 60 °C for 1 h at varying pH, then subjected to LC/MS analysis. The conversion rate was extremely high across pH 5–9, with slight alkaline conditions optimal for derivatization (Figure 2a, Appendix A). For known derivatization reagents, the pH conditions for AA derivatization were normally between 8 and 9 [26,27,28]. Thus, this method fits a wider range of pH conditions (pH 5–9) for AA derivatization, covering physiological conditions (pH 7.4).

Further, we investigated the effect of temperature on the derivatization procedure. The reaction mixtures were incubated at pH 9 for 2 days at varying temperatures and then subjected to LC/MS analysis. The results showed that the yields of Ala and Phe derivatization increased with elevated temperatures (Figure 2b, Appendix A). At 37 °C and 50 °C, the conversion rate of the carbamoyl-Ala was relatively low. When the temperature exceeded 60 °C, the reaction conversion rate improved qualitatively. This derivatization reaction is mainly thermodynamically catalyzed, so the higher the temperature, the faster the reaction rate. At 80 °C, the derivatization efficiency of the carbamoyl-Ala process was approximately 34 times that at 37 °C, and approximately 2 times that at 70 °C. Given that the temperatures utilized in current AA derivatization methods are very wide, ranging from 6 °C to 90 °C [25,29,30,31,32,33], the optimal temperature of this method is still in an appropriate range.

Then, we investigated the ideal reaction time. The conversion of raw AAs was nearly complete after 4 h of reaction at pH 9 and 80 °C (Figure 2c, Appendix A). This demonstrates that the derivatization process can be completed in 4 h, and the derivatization process only requires adding urea to the sample, adjusting the pH, and heating the sample to achieve quantitative analysis. Compared with the AA content detection kits on the market, urea does not seem to be particularly advantageous as a derivatization reagent in terms of derivatization time. However, the method is highly promising for AA derivatization because of its ease of operation, simple steps, and use of a common and inexpensive derivatization reagent.

### 2.3. Derivatization of 20 Amino Acid Mixtures and Real Samples

Derivatization of a mixture of 20 common AAs (Ala, Phe, Arg, Lys, Ser, Thr, His, Leu, Ile, Met, Val, Pro, Gly, Trp, Cys, Tyr, Glu, Gln, Asp, and Asn) yields the corresponding derivatized product, CAAs, which could be efficiently separated and identified on LC-MS. The extracted ion chromatograms were acquired using both the non-derivatization and derivatization methods under the same LC-MS conditions. Most of the non-derivatized AAs were eluted within 5 min and all the non-derivatized AAs were eluted from the column within 15 min (Figure 3a). The derivatized products, CAAs, were gradually eluted from 2 to 21 min and showed good separation, narrower than their corresponding AAs, which helps improve the separation effect and quantitative accuracy. The block of positively charged NH_2_ in AAs by urea might effectively overcome the trailing of AAs. The retention time of each CAA was extended compared to the precursor AA (Figure 3c, Appendix A). The conversion rates of all 20 CAAs were above 96%, which was favorable for the quantitative detection of AAs (Appendix A). It is noted that this reaction created two unique CAAs: disulfide-bonded carbamoyl cysteine dimer 3,3′-disulfanediylbis (2-ureidopropanoic acid) (Appendix A) and N^2^, N^6^-dicarbamoyllysine (Appendix A).

Biological samples are more complex than standard AA mixtures. To evaluate the discriminative capacity of our scheme for complex samples, we selected cell culture media DMEM and RPMI 1640 as models, which contain 15 and 19 common AAs, respectively, together with various components such as vitamins and inorganic salts (Appendix A). The concentrations of different AAs in complex samples varied widely, ranging from 5 to 584 mg/L. Under optimized reaction conditions, all AAs were transformed into CAAs with nearly 100% yield (Appendix A). As expected, the resolution of AAs improved after derivatization during LC analysis. In the control sample of DMEM (Figure 4a), eight AAs were eluted from the reversed-phase column within 2.5–3.5 min, and the ion current peaks were essentially overlapping. After derivatization (Figure 4b), this situation was significantly improved. Most AAs (Lys, His, Gly, Thr, Val, Met, Ile, Leu, Tyr, Phe, and Trp) could achieve baseline separation. The analysis of RPMI 1640 gave similar results (Appendix A). The dimer 3,3′-disulfanediylbis and N2, N6-dicarbamoyllysine mentioned above were also observed in complex samples.

### 2.4. Derivatization of Peptides

The preceding investigations developed a relatively ideal process for AA derivatization. We asked whether the method could be extended to peptide derivatization. Two dipeptides and one tripeptide (Ala-Ala, Phe-Phe, and Ala-Gly-Pro) were chosen as models to verify it. To our delight, both dipeptide and tripeptide can be quantitatively derivatized by urea (Figure 5a,b, and Appendix A) and result in an increase in UV response and a decrease in polarity, achieving similar outcomes to AA derivatization.

## 3. Materials and Methods

### 3.1. Materials

Urea was purchased from Solaribio (Beijing, China). The L-AAs discussed in our work were obtained from Macklin (Shanghai, China), including alanine (Ala), phenylalanine (Phe), glycine (Gly), proline (Pro), lysine (Lys), cysteine (Cys), serine (Ser), histidine (His), arginine (Arg), leucine (Leu), isoleucine (Ile), valine (Val), methionine acid (Met), aspartic acid (Asp), asparagine (Asn), glutamic acid (Glu), glutamine (Gln), tryptophan (Trp), threonine (Thr), and tyrosine (Tyr). Peptides were synthesized by Anhui Guoping Pharmaceutical Co., Ltd. (Anqing, China), including alanyl-alanine (Ala-Ala), phenylalanyl-phenylalanine (Phe-Phe), and alanyl-glycyl-proline (Ala-Gly-Pro). Cell culture media DMEM and RPMI 1640 were purchased from Corning (USA).

### 3.2. Preparation of Standard Curve

A total of 1 μM, 10 μM, 100 μM, 1 mM, and 10 mM of L-Ala or L-Phe, prepared from serial dilution, were incubated with 5 M urea in an aqueous solution for 2 d at 60 °C and pH 9. The reaction mixture with a total volume of 1 mL was incubated in a 1.5 mL Eppendorf tube. The resulting mixtures were subjected to LC-MS analysis. Each concentration point was conducted for three biological replicates. The response of the instrument (peak area) was plotted against the concentration of the calibration levels to give the standard curve. A linear regression model was applied to fit the data points, and the goodness of fit was evaluated using the correlation coefficient (R^2^). Within the linear range, R^2^ should be greater than 0.99.

LOD and LOQ were calculated using the signal-to-noise ratio (S/N). LOD = 3 × (S/N); LOQ = 10 × (S/N). S/N values were obtained from the mass spectrometry data analysis software Xcalibur (v.4.1) (Thermo Scientific) and were calculated automatically when the peak area was integrated into the software.

### 3.3. Derivatization Condition Optimization

To optimize derivatization conditions, 10 mM L-Ala or L-Phe were incubated with 5 mM urea in an aqueous solution under varying conditions. The reaction mixture with a total volume of 1 mL was incubated in a 1.5 mL Eppendorf tube. For evaluation of the pH effect on urea derivatization, reaction conditions were T = 60 °C, t = 2 d, and pH = 3, 5, 7, 9, or 11. For the evaluation of the temperature (T) effect on urea derivatization, reaction conditions were pH = 9, t = 2 d, and T = 37 °C, 50 °C, 60 °C, 70 °C, or 80 °C. For the evaluation of the reaction time (t) effect on urea derivatization, reaction conditions were pH = 9, T = 60 °C, and t = 1 h, 2 h, 2.5 h, 3 h, 4 h, 5 h, or 8 h. Each condition was conducted for three biological replicates.

### 3.4. Derivatization Reaction with AA Mixture, Real Samples, and Peptides

An AA mixture containing 20 different AAs with L-Lys, L-His, L-Arg, L-Gly, L-Ser, L-Asn, L-Ala, L-Gln, L-Asp, L-Thr, L-Glu, L-Cys, L-Pro, L-Val, L-Met, L-Ile, L-Leu, L-Tyr, L-Phe, and L-Trp was prepared in advance. The AA mixture (10 mM of each AA) was incubated with 5 M Urea at 80 °C and pH 9 for 4 h. The reaction mixture with a total volume of 1 mL was incubated in a 1.5 mL Eppendorf tube.

The cell culture media DMEM and RPMI 1640 were selected as complex samples. Their derivatization reactions were performed under the same conditions as for the AA mixture.

For peptide derivatization, 10 mM dipeptides or 20 mM tripeptides were incubated with 5 M urea in an aqueous solution for 4 h at pH 9 and 80 °C. The reaction mixture with a total volume of 1 mL was incubated in a 1.5 mL Eppendorf tube.

### 3.5. LC-ESI-MS Analysis

Prior to HPLC-MS analysis, the reaction mixture was filtered through a 0.22 μm aqueous polyethersulfone needle filter (Anpel) using a 1 mL disposable syringe. Then, 8 μL of the filtrate was subjected to analysis.

The HPLC was performed on the UltiMate 3000 HPLC system with an Agilent TC-C18 (2), 250 × 4.6 mm reversed-phase column. HPLC conditions for a single AA derivative were as follows: (1) mobile phase A: deionized water (0.1% formic acid); mobile phase B: acetonitrile. The gradient program was as follows: 0–5 min, 5%; 5–13 min, 5–75%; 13–15 min, 75%; 15–17 min, 75–5%, 17–22 min, 5% B. The eluent flow rate was 1 mL/min, the column was maintained at 30 °C, and 8 μL of the sample was injected. (2) Mobile phase A: deionized water (0.1% formic acid); mobile phase B: acetonitrile. The gradient program was as follows: 0–10 min, 50% B. The eluent flow rate was 1 mL/min and the column was maintained at 30 °C and 2 μL of the sample was injected (this set was only used for the optimization of the derivatization time for Phe).

HPLC conditions for mixed derivatives of 20 AAs were as follows: mobile phase A: deionized water (0.1% formic acid); mobile phase B: acetonitrile. The gradient program was as follows: 0–5 min, 5%; 5–45 min, 5–90%; 45–50 min, 90–5%; 50–55 min, 5% B. The eluent flow rate was 1 mL/min and the column was maintained at 30 °C and 8 μL of the sample was injected.

HPLC conditions for real samples were as follows: mobile phase A: deionized water (0.1% formic acid); mobile phase B: acetonitrile. The gradient program was as follows: 0–5 min, 0.5%; 5–45 min, 0.5–90%; 45–50 min, 90–0.5%; 50–55 min, 0.5% B. The eluent flow rate was 1 mL/min and the column was maintained at 30 °C and 8 μL of the sample was injected.

For all measurements, the wavelength for UV detection was 210 nm. MS and MS_2_ were performed on a Q-Exactive Plus system in positive mode. MS parameters were as follows: spray voltage of 3.8 kV, capillary temperature 320 °C, sheath gas flow rate of 3 L/min, positive mode, and scan range m/z 50–750. The data were analyzed using Xcalibur (Thermo Scientific).

## 4. Conclusions

In this work, we successfully exploited urea as a derivatization reagent for AA analysis. Twenty common AAs were shown to be quantitatively modified by urea, then successfully separated and determined in LC-MS. Each of the derivatized products, CAAs, has a longer retention time on the column compared to the precursor AA. In addition, this derivatization method is also applicable to the derivatization of short peptides. Due to the wide pH compatibility and good water solubility of urea, the derivatization reaction does not need any buffering or organic solvent involved in available derivatization reagents. The derivatization reagent, urea, is cheap, stable in aqueous environments, safe, and non-polluting. The practical method for AA analysis by LC-MS using precolumn derivatization with urea can complement the existing derivatization reagents in some aspects and has broad application potential in the future. It could be easily extended to other analytical methods or applications by utilizing urea attached to a functional moiety such as a fluorophore, fluorinated group, or the chiral recognition group.

## Figures and Tables

**Figure 1 ijms-24-07332-f001:**
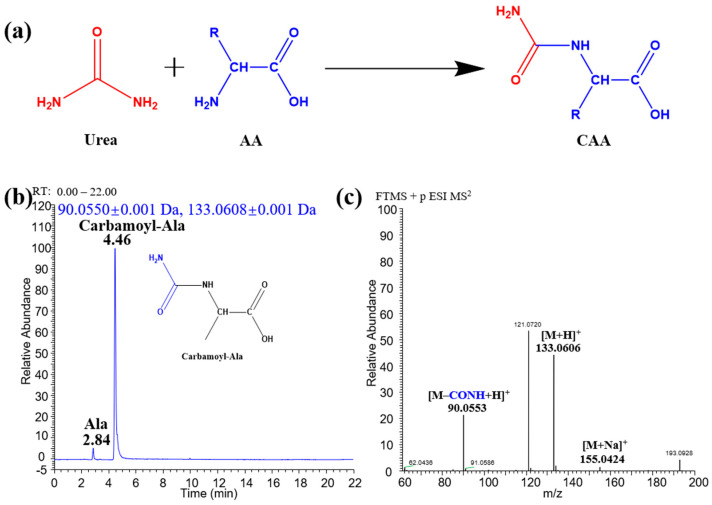
Urea derivatization reaction with AAs. (**a**) Reaction scheme. (**b**) EIC-MS profile of Ala and carbamoyl-Ala. (**c**) MS/MS spectrum of the product carbamoyl-Ala.

**Figure 2 ijms-24-07332-f002:**
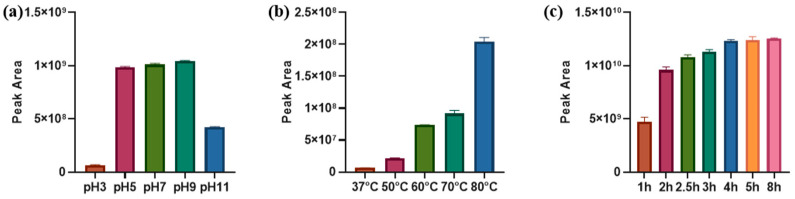
Effects of pH, temperature, and time effect on urea derivatization. (**a**) The conversion rate of carbamoyl-Ala under different pH conditions (3–11). (**b**) The conversion rate of carbamoyl-Ala under different temperature conditions (37 °C–80 °C). (**c**) The conversion rate of carbamoyl-Ala under different time conditions (1–8 h). Each condition was conducted for three biological replicates. HPLC method of Figure 2: mobile phase A: deionized water (0.1% formic acid); mobile phase B: acetonitrile. The gradient program was as follows: 0–5 min, 5%; 5–13 min, 5–75%; 13–15 min, 75%; 15–17 min, 75–5%, 17–22 min, 5% B. The eluent flow rate was 1 mL/min and the column was maintained at 30 °C and 8 μL of the sample was injected.

**Figure 3 ijms-24-07332-f003:**
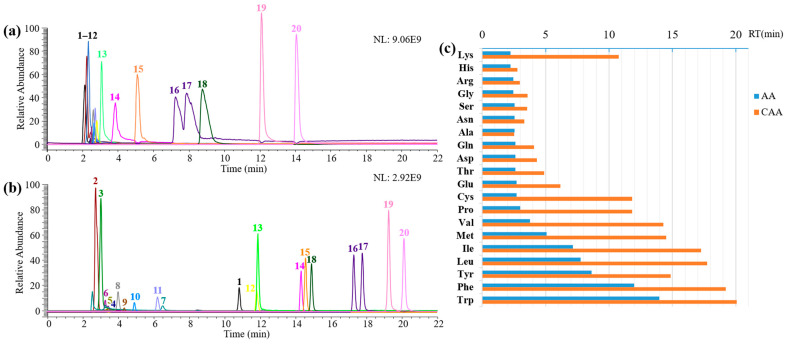
Derivatization of 20 AA Mixtures. (**a**) EIC-MS profile of 20 AAs. (**b**) EIC-MS profile of 20 CAAs. Unmarked peaks are unreacted AA and unidentified derivatization byproducts. (**c**) Changes in AA retention time on reversed-phase C18 column before and after derivatization: orange (AA) and blue (CAA). The abscissa is the retention time in minutes, and the ordinate is the AA. AAs are marked with numbers as follows, 1: Lys, 2: His, 3: Arg, 4: Gly, 5: Ser, 6: Asn, 7: Ala, 8: Gln, 9: Asp, 10: Thr, 11: Glu, 12: Cys, 13: Pro, 14: Val, 15: Met, 16: Ile, 17: Leu, 18: Tyr, 19: Phe and 20: Trp. The m/z extraction range of AAs and CAAs were listed in Appendix A.

**Figure 4 ijms-24-07332-f004:**
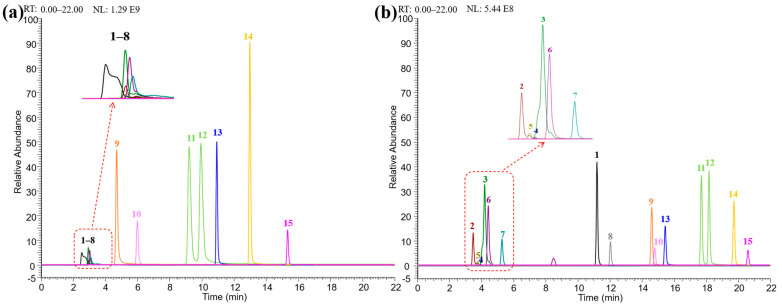
Derivatization of a complex sample. (**a**) EIC-MS profile of AAs in cell culture medium DMEM. (**b**) EIC-MS profile of CAAs in cell culture medium DMEM after derivatization. AAs are marked with numbers as follows, 1: Lys, 2: His, 3: Arg, 4: Gly, 5: Ser, 6: Gln, 7: Thr, 8: Cys, 9: Val, 10: Met, 11: Ile, 12: Leu, 13: Tyr, 14: Phe and 15: Trp. The m/z extraction range of AAs and CAAs are listed in Appendix A.

**Figure 5 ijms-24-07332-f005:**
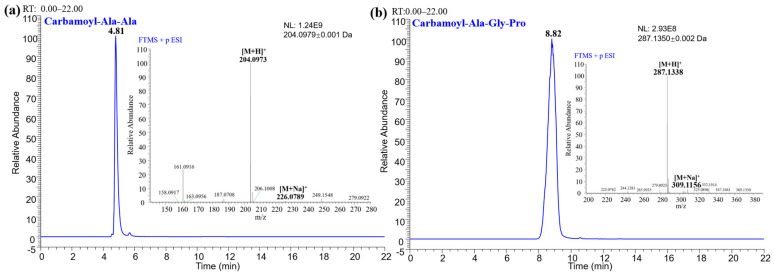
Derivatization of Peptides. (**a**) EIC-MS profile of carbamoyl-Ala-Ala. (**b**) EIC-MS profile of carbamoyl-Ala-Gly-Pro.

**Table 1 ijms-24-07332-t001:** Validation results of the LC-MS/MS method for the analysis of derivatized AAs.

Analyte	Linear Equation	Linear Range (nM)	R^2^	LOD (nM) ^a^	LOD (nM) ^b^	LOQ (nM) ^c^	LOQ (nM) ^d^
Ala	y = 7.5905 × 10^11^x + 672444175.81	10^3^–10^7^	0.9939	130.78	58.60	435.50	195.40
Phe	y =1.10681 × 10^12^x + 1634677812.7	10^3^–10^7^	0.9923	19.96	11.64	66.47	38.80

^a^: LOD of AA; ^b^: LOD of CAA; ^c^: LOQ of AA; ^d^: LOQ of CAA. LOD and LOQ are calculated using the signal-to-noise ratio (S/N). LOD = 3 × (S/N); LOQ = 10 × (S/N). S/N values are obtained from the mass spectrometry data analysis software Xcalibur (v.4.1) (Thermo Scientific, Waltham, MA, USA) and are calculated automatically when the peak area is integrated into the software.

## Data Availability

The data is contained within the article.

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
