# Peer review of "A Practical Method for Amino Acid Analysis by LC-MS Using Precolumn Derivatization with Urea"

_ijms, 2023, doi:10.3390/ijms24087332_

Round 1

Reviewer 1 Report

The results obtained are interesting from a scientific and practical point of view.

I wonder what is the stability of derivatives over time.

Author Response

Point 1: The results obtained are interesting from a scientific and practical point of view. I wonder what is the stability of derivatives over time.

Response: Thank you for your comments. The urea-derivatives of 20 AAs are stable at 4℃ within 6 months. Stability over longer periods of time has not been determined.

Reviewer 2 Report

The article is devoted to the pre-column modification of amino acids with a urea residue. The article is written in good language and sounds scientifically. The advantages of this method are an increase in the retention times of carbomoyl derivatives of amino acids and oligopeptides, a decrease in peak width and almost complete separation at the baseline level. This also leads to the possibility of detecting amino acids with a simple UV detector without resorting to more complex refractometric or light scattering detectors. Another advantage is the use of a simple, cheap, and non-toxic derivatization reagent - urea.

The disadvantages of the described method include the use of rather strong heating (80°C), which can lead to the decomposition of analytes in complex matrices, as well as a long derivatization time (4 hours), which makes this method difficult to apply in routine studies. Also, this method is practically useless in the presence of LCMS equipment due to the EIC mode and the use of HILIC columns.

The effectiveness of the technique for detecting amino acids and oligopeptides in complex mixtures can be considered proven. However, there is insufficient evidence for the applicability of this method for quantitation. Some study is given for Ala, c-Cis and N2,N6-dicarbamoyllysine only.

This article may be of interest to researchers and scientists working in the field of analysis of natural compounds due to the ubiquity of instruments with a typical UV detector.

Reviewer notes

It would be convenient to see representative UV spectra of amino acids and their derivatives in the article.

Page 1, bottom "In recent years, numerous HPLC methods capable of UV/FL detection have been developed for the analysis of AAs." References must be provided.

If follow the reference 22, then there is no mention of urea as a modifying agent. It is necessary to correctly describe the meaning of this link in the proper place of the article.

Table 1 contains linear regression equations; they should be formatted for better reading. The plus sign to the power of a floating-point number (argument x) is confusing and looks like the "addition" sign in an equation. For example, you can set the multiplication sign in front of x - ×x or *x.

The LOD and LOQ for non-derivatized Ala and Phe should be revealed in the article as evidence of less advanced detection than the derivatized ones. It is necessary to describe the methodology for determining the above parameters for clarity. It is not clear from the main article and SI whether the derivatization of the complex mixture of AAs was quantitative, nor the post-derivatization EIC-MS chromatograms of the original amino acids are shown.

Chapter 3.2. - incubation conditions are unclear.

Page 2 “Especially with the introduction of MS acquiring strategies like multiple reaction monitoring (SRM) [11-13] and selected reaction monitoring (MRM)” MRM and SRM abbreviations must be swapped, typos.

EIC-MS profiles must be provided with the digital legend like 90.0550±0.002 Da everywhere (main article and SI).

Figure S15. “EIC-MS profile of disulfide-linked amino group formyl-cysteine dimer”: Please check substance naming.

Chapter 3.4., paragraph 2. “The eluent flow rate was 1 mL /min and the column was maintained at 30 ℃and 8 μL of the sample was injected.” Space is missing.

Conclusion

Accept after minor revision.

Author Response

Point 1: It would be convenient to see representative UV spectra of amino acids and their derivatives in the article.

Response: Thank you for your reminder. we have added the UV spectra (210 nm) to the supporting information (Figure S5 and Figure S6).

Point 2: Page 1, bottom "In recent years, numerous HPLC methods capable of UV/FL detection have been developed for the analysis of AAs." References must be provided.

Response: Thanks for your suggestion. We have added references 10-12 to the sentence:

  1. Wang, H.; McNeil, Y. R.; Yeo, T. W.; Anstey, N. M. Simultaneous determination of multiple amino acids in plasma in critical illness by high performance liquid chromatography with ultraviolet and fluorescence detection. J Chromatogr B: Anal. Technol. Biomed. Life Sci. 2013, 940, 53-8.

11.Song, Y.; Funatsu, T.; Tsunoda, M. Amino acid analysis using core-shell particle column. J. Chromatogr. B: Anal. Technol. Biomed. Life Sci. 2013, 927, 214-7.

12.Tasakis, R. N.; Touraki, M. Identification of bacteriocins secreted by the probiotic Lactococcus lactis following microwave-assisted acid hydrolysis (MAAH), amino acid content analysis, and bioinformatics. Anal. Bioanal. Chem. 2018, 410 (4), 1299-1310.

Point 3: If follow the reference 22, then there is no mention of urea as a modifying agent. It is necessary to correctly describe the meaning of this link in the proper place of the article.

Response: Thanks for your reminder. We retained the reference 22 on page 2 and page 4 (now reference 25), and deleted the reference to original reference 22 in chapter 2.1 (because it is inappropriate).

Point 4: Table 1 contains linear regression equations; they should be formatted for better reading. The plus sign to the power of a floating-point number (argument x) is confusing and looks like the "addition" sign in an equation. For example, you can set the multiplication sign in front of x - ×x or *x.

Response: Thanks for your kind suggestion. We have modified the equations in Table 1.

Point 5: The LOD and LOQ for non-derivatized Ala and Phe should be revealed in the article as evidence of less advanced detection than the derivatized ones. It is necessary to describe the methodology for determining the above parameters for clarity. It is not clear from the main article and SI whether the derivatization of the complex mixture of AAs was quantitative, nor the post-derivatization EIC-MS chromatograms of the original amino acids are shown.

Response: thanks for your comments to improve our manuscript.

The LOD and LOQ for non-derivatized Ala and Phe have been added to Table 1, and the descriptions of LOD and LOQ determination have been added to the legend of Table 1 and materials and methods section.

  • We supplemented the AA conversion rate data, including 20 AA mixture (Table S3) and cell-culture medium sample (Table S6). These data showed that all the conversion rate were > 96%, which suggest the derivatization of the complex mixture of AAs was almost quantitative. Meanwhile, we also extracted the EIC-MS chromatograms of the original amino acids in post-derivatization EIC-MS spectra (Figure S16).

Point 6: Chapter 3.2. - incubation conditions are unclear.

Response: Thanks for your reminder. We have added detailed descriptions of experimental procedures.

Point 7: Page 2 “Especially with the introduction of MS acquiring strategies like multiple reaction monitoring (SRM) [11-13] and selected reaction monitoring (MRM)” MRM and SRM abbreviations must be swapped, typos.

Response: Thanks for your kind reminder. We have corrected the abbreviation error.

Point 8: EIC-MS profiles must be provided with the digital legend like 90.0550±0.002 Da everywhere (main article and SI).

Response: Thanks for your comment. We have made modifications for all EIC-MS profiles.

Point 9: Figure S15. “EIC-MS profile of disulfide-linked amino group formyl-cysteine dimer”: Please check substance naming.

Response: Thanks for your comment. The revised title is“EIC-MS profile of 3,3'-disulfanediylbis (2-ureidopropanoic acid)”. The name of the substance was generated from chemdraw.

Point 10: Chapter 3.4., paragraph 2. “The eluent flow rate was 1 mL /min and the column was maintained at 30 ℃and 8 μL of the sample was injected.” Space is missing.

Response: Thanks for your kind reminder. We have corrected it.

Reviewer 3 Report

The authors present the optimization of a method for amino acids (AAs) derivatization and its application for LC/MS analysis. At first, the principals of AA derivatization applying urea are presented (reaction schemes, characterization of reaction products, effect of derivatization on LC/MS analysis). Secondly, different experimental conditions (pH, T, t) for conducting the derivatization are evaluated and optimized. Thirdly, the optimized method is applied for analysis of mixtures of AAs and small model peptides.

Some aspects of the presented work are a valuable addition to existing derivatization methods for AA analysis. Especially, the optimization of experimental derivatization conditions is rather comprehensive. However, some important analytical aspects are mixed up and are partially explained or discussed only.

Major Comments:

The application of urea for AA derivatization is not “novel” as stated in the title of the manuscript (see Ref. 24). As described in the introduction of the manuscript (last paragraph) the present work describes the experimental optimization of AA derivatization and its and application to LC/MS analysis. The title is misleading and should be changed accordingly.

The improved UV-detectability after derivatization is mentioned at several positions in the manuscript (e.g. p. 3, l. 3 ff). However, no experimental data or figures (chromatograms) are shown. Please include LC-UV data in the manuscript (supporting information) in order to prove the statements.

Table 1 includes data on quantitative analysis for Ala and Phe. There are several weak points regarding the quantitative aspect of this work: (1) The linear range, with one order of magnitude (1-10 mM), seems to be rather limited. Please verify these data. In contrast to the limited linear range, very low concentration values are given for LOQ and LOD (nM range vs mM range!?). This is contradictory and confusing. Typically, the linear range is starting at the LOQ. (2) Please specify the calibration levels and their preparation (Materials and Methods section). (3) Another important aspect for quantitative LC/MS analysis is the use of an internal standard. Was an internal standard applied for quant analysis? Especially during derivatization steps prior LC/MS analysis an (isotopically labelled) internal standard is important in order to compensate yield-variability. Please include data and discuss accordingly. (4) It would be interesting to apply the developed quantitative method on the quantitative analysis of the two cell culture media. As the AA composition including their concentrations are known (Table S2 and S3) it would be interesting to benchmark the developed method.

The conversion rates of AAs to their corresponding Carbamoyl-AAs is described to be very high (according to page 2: >98 % and > 99 % for Ala and Phe, respectively). However, as shown in Fig S3 a considerable large peak for Ala (2.84 min) is present in the EIC for Carbamoyl-Ala, indicating a conversion rate significantly lower than 98 %. This is confusing. Please clarify and improve the EIC figures depicting the conversion rates and/or include peak area data for underivatized and derivatized analysis. Please include absolute abundancies/NLs on the y-axis throughout all EICs in the manuscript and supporting info. Additionally, the explanation, how the conversion rate is calculated is misleading in the text (page 2) and needs rephrasing. A comprehensive table showing the conversion rate for all AAs studied should be included and should replace Table S4 (as it includes only 15 AAs)

The figures showing the results of the optimization experiments (Fig 2 and Fig S5) include error bars. However, no information on the number of replicates is given in the manuscript. Please complete accordingly. For the data shown in Fig 2b no info on applied pH and derivatization time is given. Please include.

In Figures S6, S7, S9, S10, S12, S13: EIC profiles of Carbamoyl-Ala and Carbamoyl-Phe are shown. Please include also the ion traces of the non-derivatized AAs in the chromatograms, showing their peak-area decrease at optimized conditions. This would improve the quality of presentation. Why an isocratic separation was applied in Fig S13?

The retention times of some AAs in Figure 4a and 4b (e.g. Trp: ~15.5 min and ~ 21 min, respectively) are shifted in comparison to Figure 3a and 3b (Trp: ~14 min and ~ 20 min, respectively). Similar shifts are observed in Figure S17. Please correct/explain/discuss the shifted retention times observed for analysis of the cell culture media in comparison to the 20 AA mixture.

Minor comments:

page 1 line 2 ff: please use the introduced abbreviation “AAs“ throughout the whole manuscript

p. 1 l. 6 f: “AAs are essential…” Please check meaning and language

p. 1 l. 12: “The AA detectors are mainly based…” instead of “The AA detectors mainly based…”

p 2 l. 1: “…by-products…” instead of “by-product”

p. 2 l. 2f: please rephrase the sentence “With the rapid development….”

p2. l.10/11: the terms “SRM” and “MRM” are mixed up, please correct

p. 3 l. 2: The increase of retention time for Ala and Carbamoyl-Ala is 1.62 min (and not 1.61 min), according to Fig. 1b. Please correct.

p. 5 l. 3: “…samples” instead of “sample” and “mixtures” instead of “mixture”

Figure S14 is redundant to Fig 1b (why retention times slightly shifted?) and can therefore be removed.

Please include detailed experimental information on the derivatization reaction: volumes of solutions, tubes, dilution/pH adjustment prior LC analysis

Supporting information: p.1: Please remove “Analytical and Bioanalytical Chemistry” in the title.

Author Response

Point 1: The application of urea for AA derivatization is not “novel” as stated in the title of the manuscript (see Ref. 24). As described in the introduction of the manuscript (last paragraph) the present work describes the experimental optimization of AA derivatization and its and application to LC/MS analysis. The title is misleading and should be changed accordingly.

Response: Thanks for your comment. Considering that the most important feature of this method is practical (namely, easy-handled, highly tolerant, cheap, etc.), we decided to change the title to “A practical method for amino acid analysis by LC-MS using precolumn derivatization with urea”.

Point 2: The improved UV-detectability after derivatization is mentioned at several positions in the manuscript (e.g. p. 3, l. 3 ff). However, no experimental data or figures (chromatograms) are shown. Please include LC-UV data in the manuscript (supporting information) in order to prove the statements.

Response: Thank you for your reminder. we have added the UV spectra of alanine UV detection (210 nm) to the supporting information (Figure S5 and Figure S6).

Point 3: Table 1 includes data on quantitative analysis for Ala and Phe. There are several weak points regarding the quantitative aspect of this work: (1) The linear range, with one order of magnitude (1-10 mM), seems to be rather limited. Please verify these data. In contrast to the limited linear range, very low concentration values are given for LOQ and LOD (nM range vs mM range!?). This is contradictory and confusing. Typically, the linear range is starting at the LOQ. (2) Please specify the calibration levels and their preparation (Materials and Methods section). (3) Another important aspect for quantitative LC/MS analysis is the use of an internal standard. Was an internal standard applied for quant analysis? Especially during derivatization steps prior LC/MS analysis an (isotopically labelled) internal standard is important in order to compensate yield-variability. Please include data and discuss accordingly. (4) It would be interesting to apply the developed quantitative method on the quantitative analysis of the two cell culture media. As the AA composition including their concentrations are known (Table S2 and S3) it would be interesting to benchmark the developed method.

Response: Thank you for your comments.

  • We have to apologize for our careless on the concentration unit. The linear range was 1 μM-10 mM instead of 1-10 mM. We initially tested this concentration range (1 μM-10 mM), which covers the concentration of most biological samples, and found that it has good linearity. We didn't study a wider concentration range.
  • The descriptions have been added to chapter 3.2.
  • Thank you very much for pointing out this important issue. Unfortunately, due to the limited time, we did not supplement experimental validation. The key motivation of this study is to demonstrate the feasibility and general advantages of urea derivatization for AA analysis. Experimentally, label free based relative quantitative method was utilized without the introduction of internal standard, which may lack a bit of quantitative accuracy, but didn’t affect the main conclusion.
  • Thank you for your kind suggestion. The presentation of this data requires the preparation of the calibration curve and the analysis of the analyte in the same batch. Due to the limitations of time, we did not supplement experimental validation.

Point 4: The conversion rates of AAs to their corresponding Carbamoyl-AAs is described to be very high (according to page 2: >98 % and > 99 % for Ala and Phe, respectively). However, as shown in Fig S3 a considerable large peak for Ala (2.84 min) is present in the EIC for Carbamoyl-Ala, indicating a conversion rate significantly lower than 98 %. This is confusing. Please clarify and improve the EIC figures depicting the conversion rates and/or include peak area data for underivatized and derivatized analysis. Please include absolute abundancies/NLs on the y-axis throughout all EICs in the manuscript and supporting info. Additionally, the explanation, how the conversion rate is calculated is misleading in the text (page 2) and needs rephrasing. A comprehensive table showing the conversion rate for all AAs studied should be included and should replace Table S4 (as it includes only 15 AAs)

Response: Thank you for your great comments.

  • We have added NLs to all those EIC figures. In Figures S3 and S4, we calculated the integration and conversion of amino acid peaks, which showing that the conversion rates of the two AAs indeed >98 %.
  • The explanation of conversion rate calculation was added to page 2.
  • As there are only 15 AAs in Cell culture medium DMEM (Table S6), we have presented all of them in Table S3. In addition, we also added the conversion rate of 20 standard AA mixtures (Table S6).

Point 5: The figures showing the results of the optimization experiments (Fig 2 and Fig S5) include error bars. However, no information on the number of replicates is given in the manuscript. Please complete accordingly. For the data shown in Fig 2b no info on applied pH and derivatization time is given. Please include.

Response: Thank you for your reminder.

  • The descriptions of replication have added to figure legends (of Fig 2 and Fig S5) and “Materials and Methods” section.
  • Applied pH and derivatization time corresponding to Fig 2b have added to “Materials and Methods” section.

Point 6: In Figures S6, S7, S9, S10, S12, S13: EIC profiles of Carbamoyl-Ala and Carbamoyl-Phe are shown. Please include also the ion traces of the non-derivatized AAs in the chromatograms, showing their peak-area decrease at optimized conditions. This would improve the quality of presentation. Why an isocratic separation was applied in Fig S13?

Response: Thank you for your comments.

  • The ion traces of the non-derivatized AAs in the chromatograms have been presented.
  • For Fig S13, we apologize for the confusing.The isocratic separation for Phe was designed for UV detection, as the 22 min gradient resulting the irregular peak shape of UV spectra of phe and carbamoyl-Phe. The similar experiments were also done using 22 min analysis but lack some time points. Generally, the LC gradient of the analysis will not affect the relative quantitative results as long as it enables effective separation, such that we presented this isocratic separation analysis results

Point 7: The retention times of some AAs in Figure 4a and 4b (e.g. Trp: ~15.5 min and ~ 21 min, respectively) are shifted in comparison to Figure 3a and 3b (Trp: ~14 min and ~ 20 min, respectively). Similar shifts are observed in Figure S17. Please correct/explain/discuss the shifted retention times observed for analysis of the cell culture media in comparison to the 20 AA mixture.

Response: Thank you for your comment.

Considering the presence of some impurities in the culture-medium samples, such as salt and vitamins, we modified the LC gradient to achieve better analysis (see below). The descriptions have been also added to “Materials and Methods” section.

HPLC conditions for 20 AA mixture were as follows: mobile phase A: deionized water (0.1% formic acid); mobile phase B: acetonitrile. Gradient program was as follows: 0-5 min, 5%; 5-45 min, 5-90%; 45-50 min, 90-5%; 50-55min, 5% B. The eluent flow rate was 1 mL/min and the column was maintained at 30 °C and 8 μL of the sample was injected.

HPLC conditions for mixed derivatives of real samples (cell culture medium DMEM and RPMI 1640) as follows: mobile phase A: deionized water (0.1% formic acid); mobile phase B: acetonitrile. Gradient program was as follows: 0-5 min, 0.5%; 5-45 min, 0.5-90%; 45-50 min, 90-0.5%; 50-55min, 0.5% B. The eluent flow rate was 1 mL/min and the column was maintained at 30 °C and 8 μL of the sample was injected.

Point 8: Minor comments:

page 1 line 2 ff: please use the introduced abbreviation “AAs“ throughout the whole manuscript

  1. 1 l. 6 f: “AAs are essential…” Please check meaning and language
  2. 1 l. 12: “The AA detectors are mainly based…” instead of “The AA detectors mainly based…”

p 2 l. 1: “…by-products…” instead of “by-product”

  1. 2 l. 2f: please rephrase the sentence “With the rapid development….”

p2. l.10/11: the terms “SRM” and “MRM” are mixed up, please correct

  1. 3 l. 2: The increase of retention time for Ala and Carbamoyl-Ala is 1.62 min (and not 1.61 min), according to Fig. 1b. Please correct.
  2. 5 l. 3: “…samples” instead of “sample” and “mixtures” instead of “mixture”

Figure S14 is redundant to Fig 1b (why retention times slightly shifted?) and can therefore be removed.

Please include detailed experimental information on the derivatization reaction: volumes of solutions, tubes, dilution/pH adjustment prior LC analysis

Supporting information: p.1: Please remove “Analytical and Bioanalytical Chemistry” in the title.

Response: Thank you for your careful guidance.

In full accordance with your valuable comments, we have revised the manuscript. The revisions were marked in yellow.

Round 2

Reviewer 3 Report

Major:

For Ala the LCUV data shown is ok (RT for Ala 2.9 min and for Carbamoyl-Ala 4.4 min, respectively, in LCMS and LCUV experiment, as shown in Fig. S3 and S5). For Phe the LCUV data is confusing to the reader (Fig. S6): The signal intensity of carbamoyl-Phe is not increased by a factor of 1.26 as mentioned in the manuscript (page 3, line 4). The signal increase seems to be higher. However, the peak shape of carbamoyl-Phe in the LCUV chromatogram (Fig. S6) is very poor in comparison to the LCMS experiment (Fig. S4). This is due to differing chromatographic conditions. The same chromatographic conditions as applied in the LCMS experiment (Fig. S4) should be applied for UV detection (as shown for Ala). Please change the chromatographic data accordingly.

Minor:

Conversion rates calculation: “x100” is missing in Fig. S3 und S4 legends.

Fig. S8 and S9, S11, S12, S14 legends: “Chromatographic conditions same as in Fig. S6”? Please check chromatographic conditions (must be “same as Fig. S5”).

Please include detailed experimental information in the material and methods section regarding the derivatization: on (i) applied volumes of solutions, (ii) applied tubes, (iii) dilution/pH adjustment prior LC analysis.

Author Response

Point 1: For Ala the LCUV data shown is ok (RT for Ala 2.9 min and for Carbamoyl-Ala 4.4 min, respectively, in LCMS and LCUV experiment, as shown in Fig. S3 and S5). For Phe the LCUV data is confusing to the reader (Fig. S6): The signal intensity of carbamoyl-Phe is not increased by a factor of 1.26 as mentioned in the manuscript (page 3, line 4). The signal increase seems to be higher. However, the peak shape of carbamoyl-Phe in the LCUV chromatogram (Fig. S6) is very poor in comparison to the LCMS experiment (Fig. S4). This is due to differing chromatographic conditions. The same chromatographic conditions as applied in the LCMS experiment (Fig. S4) should be applied for UV detection (as shown for Ala). Please change the chromatographic data accordingly.

Response: Thank you for your comments.

We have provided LCUV chromatogram spectrum (new Fig. S6) which was prepared from the same chromatographic conditions as applied in the LCMS experiment (Fig. S4).

As you can see, UV absorption of Phe (RT=9.26 min; new Fig. S6) is just overlapped with the solvent absorption peak, which may affect quantification. That’s why we performed a LCUV run with a 10 min isocratic separation.

Point 2: Conversion rates calculation: “x100” is missing in Fig. S3 und S4 legends.

Fig. S8 and S9, S11, S12, S14 legends: “Chromatographic conditions same as in Fig. S6”? Please check chromatographic conditions (must be “same as Fig. S5”).

Response: Thank you for your reminder. We have corrected these mistakes. The revisions were marked in blue.

Point 3: Please include detailed experimental information in the material and methods section regarding the derivatization: on (i) applied volumes of solutions, (ii) applied tubes, (iii) dilution/pH adjustment prior LC analysis.

Response: Thank you for your reminder. The information you mentioned have been supplemented to “Materials and Methods” section (chapter 3.2 ~ 3.5). The revisions were marked in blue.